# Vascular Growth in Lymphomas: Angiogenesis and Alternative Ways

**DOI:** 10.3390/cancers15123262

**Published:** 2023-06-20

**Authors:** Domenico Ribatti, Roberto Tamma, Tiziana Annese, Antonio d’Amati, Giuseppe Ingravallo, Giorgina Specchia

**Affiliations:** 1Department of Translational Biomedicine and Neurosciences, School of Medicine, University of Bari “Aldo Moro”, 70124 Bari, Italy; roberto.tamma@uniba.it (R.T.); antonio.damati@uniba.it (A.d.); 2Department of Medicine and Surgery, Libera Università del Mediterraneo (LUM) “Giuseppe Degennaro”, 70124 Bari, Italy; annese@lum.it; 3Section of Anatomical and Molecular Pathology, Department of Precision and Regenerative Medicine and Jonian Area, School of Medicine, University of Bari “Aldo Moro”, 70124 Bari, Italy; giuseppe.ingravallo@uniba.it; 4School of Medicine, University of Bari “Aldo Moro”, 70124 Bari, Italy; specchia.giorgina@gmail.com

**Keywords:** angiogenesis, intussusceptive microvascular growth, lymphomas, vasculogenic mimicry, vascular cooption

## Abstract

**Simple Summary:**

Angiogenesis is the main mechanism for the formation of new blood vessels in tumors. Recent studies also described alternative non-angiogenic ways that play an important role in tumoral resistance to anti-angiogenic therapies. Lymphoma cells can induce the formation of new blood vessels via both angiogenic and non-angiogenic mechanisms. Despite the numerous discoveries regarding lymphomas’ pathogenesis, less is known about these processes and their role in tumoral initiation and progression. This review discusses the current knowledge on vessel formation in lymphomas, highlighting the potential implications for prognosis and treatment.

**Abstract:**

The formation of new blood vessels is a critical process for tumor growth and may be achieved through different mechanisms. Angiogenesis represents the first described and most studied mode of vessel formation, but tumors may also use alternative ways to secure blood supply and eventually acquire resistance to anti-angiogenic treatments. These non-angiogenic mechanisms have been described more recently, including intussusceptive microvascular growth (IMG), vascular co-option, and vasculogenic mimicry. Like solid tumors, angiogenic and non-angiogenic pathways in lymphomas play a fundamental role in tumor growth and progression. In view of the relevant prognostic and therapeutic implications, a comprehensive understanding of these mechanisms is of paramount importance for improving the efficacy of treatment in patients with lymphoma. In this review, we summarize the current knowledge on angiogenic and non-angiogenic mechanisms involved in the formation of new blood vessels in Hodgkin’s and non-Hodgkin’s lymphomas.

## 1. Tumor Angiogenesis

Vasculogenesis, or the creation of capillaries from endothelial cells differentiating in situ from mesodermal cells, causes the first blood vessels to arise throughout embryonic life. In this manner, the primitive vascular plexus and the primitive heart are created [1].

The creation of capillaries from pre-existing vessels, such as capillaries and post-capillary venules, is referred to as angiogenesis [2]. This process is based on endothelial sprouting microvascular growth. The first step in angiogenesis is the local breakdown of the basement membrane enclosing the capillaries. Next, the underlying endothelial cells invade the surrounding stroma in the path of the angiogenic stimulation. A network of new blood vessels is formed because of endothelial cell migration, which is accompanied by the proliferation of endothelial cells and their arrangement into three-dimensional structures. Physiological angiogenesis only happens during certain distinct processes in adults, including the female reproductive cycle, tissue repair, and wound healing [3].

Judah Folkman first suggested the idea that angiogenesis, which is closely related to tumor growth, is correlated with microvascular density, or the number of microvessels that can be counted in a sample tumor area using antibodies that are specific for endothelial cell markers (e.g., CD31, CD34) [4]. CD31 is more expressed in mature endothelial cells, whereas CD34 is expressed in immature blood vessels. In the meantime, as concerns endothelial basement membrane, LH39 is a monoclonal antibody recognizing an epitope located at the lamina lucida of mature small veins and capillaries but not in newly formed vessels [5].

Intra-tumoral microvessel density and prognosis have been found to be positively correlated in solid tumors, according to the greater part of the research literature [4]. Numerous studies have found associations between intratumoral microvascular density, expression of angiogenic growth factors, tumor growth, and the occurrence of metastases [6]. These findings indicate that intratumoral microvascular density provides crucial information on the extent and role of tumor vasculature.

An avascular phase precedes a vascular phase in the process of tumor growth. Most tumors develop and persist in situ, devoid of angiogenesis, for a considerable amount of time before starting the angiogenic process [2]. The histopathological image characterized by a small colony of neoplastic cells that reaches a stable state before becoming invasive represents the avascular phase. In this situation, metabolites and catabolites are distributed through the surrounding tissue by simple diffusion. While cells in the deeper part of the tumor degenerate, those on the tumor’s periphery continue to grow. The production and release of angiogenic factors or a decrease in the level of endogenous angiogenic inhibitors have both been linked to the activation of the angiogenic switch [2]. Another important aspect of tumor growth is represented by tumor dormancy, occurring during tumor initiation (“local tumor dormancy”), metastatic dissemination (“metastatic dormancy”), and escape from anti-cancer therapies (“therapy-induced dormancy”). Tumor dormancy can also be distinguished in the arrest of cancer cell proliferation within the tumor mass caused by apoptosis due to poor vascularization (“angiogenic dormancy”) or by the immune response (“immune dormancy”) [7].

## 2. Alternative Ways of Tumor Vascularization

In the last thirty years, alternative modes of vascularization of tumor growth have been described. These techniques include intussusceptive microvascular growth (IMG), vascular co-option, and vasculogenic mimicry (Figure 1 and Figure 2) [8,9,10].

In IMG, the expansion of the vascular network is achieved by inserting tissue columns into the vascular lumen of pre-existing vessels [8]. IMG occurs in various tumors, including colon and breast carcinomas, melanoma, and gliomas [11,12,13,14]. IMG has several advantages over sprouting angiogenesis, including faster blood vessel formation, metabolic cost savings due to the lack of extensive endothelial cell proliferation, basement membrane degradation, and surrounding tissue invasion during sprouting angiogenesis, and less leaky capillaries as a result.

The second pathway involves cancer cells invading and occupying normal tissues to utilize pre-existing vessels, which is referred to as vascular or vessel co-option [15]. The field of vessel co-option was introduced by Pezzella and coworkers in 1997 [16]. They demonstrated that tumor growth in non-small cell lung carcinomas occurs without angiogenesis and that, in this context, cancer cells survive by using pre-existing vessels as a source of oxygen and metabolites. An example of vascular co-option is represented in Figure 1, which shows, in the outer parts of lung metastases, the conservation of the alveolar architecture. SCID mice were injected with HT1080 tumor cells to produce experimental metastases. Tumor cells penetrate the alveolar air space after extravasation and the formation of tiny interstitial colonies. Type I alveolar epithelial cells were identified by immunostaining with anti-podoplanin antibodies (green fluorescence), blood vessels were identified by immunostaining with anti-CD31 antibodies to mark endothelial cells (red fluorescence), and nuclear staining was obtained by immunostaining with TOTO-3 (blue fluorescence).

In 1999, Holash et al. reported that tumor cells co-opt pre-existing vessels and grow around them as cuffs [17]. These authors evaluated the possibility that vascular endothelial growth factor (VEGF) and angiopoietins (Angs) interact during tumor angiogenesis in a rat glioma experimental model. They demonstrated that early after tumor cells implantation, tumor vascularization was attributable to the co-option of existing blood vessels by tumor cells. By 4 weeks after tumor cells implantation, blood vessels within the core of the tumors regressed because of the destabilizing action of Ang-2 on the vessel wall. The coopted vasculature triggers an apoptotic cascade, likely by autocrine production of Ang-2, which exterminates most of the dependent tumor and causes widespread tumor death. This is the outcome of a host defensive mechanism that has been engaged. When the ratio of VEGF to Ang-2 is high, the new tumor vessels continue to grow; when it is low, the new tumor vessels contract. The interaction of VEGF and Ang-2 at the edge of the expanding tumor mass results in angiogenesis.

In human melanoma cells, in 1999, Maniotis and colleagues initially identified the third alternative route of tumor vascularization, which they named vasculogenic mimicry to underline the development of new blood vessels independently of angiogenesis [9]. “Vasculogenic” was chosen to denote the pathway’s de novo formation, and “mimicry” was chosen since the paths employed by tumor cells to convey fluid to tissues were obviously not blood vessels. Laminin 5 and matrix metalloproteinases-1, -2, and -9 (MMP-1, MMP-2, MMP-9) were significantly more expressed in highly aggressive human cutaneous melanoma cell lines when compared to less aggressive cell lines, according to a microarray gene chip analysis. Vasculogenic mimicry describes the highly aggressive cancers’ capacity to produce blood arteries made of tumor cells rather than endothelial cells. Accordingly, the development of blood vessels in tumors may be directly influenced by cancer cells [18].

Another theory is that tumor cells take the place of the endothelial cell lining, creating what is known as mosaic vasculature, in which both endothelium and tumor cells help to construct vascular tubes [13]. This study used endogenous green fluorescent protein (GFP) to label tumor cells and CD105 to identify endothelial cells. It showed that 15% of the colon carcinoma xenografts’ perfused vasculature were made up of mosaic vessels.

## 3. Angiogenesis and Microvascular Density in Hodgkin’s Lymphomas

Lymphomas constitute a large group of lymphoproliferative disorders. Hodgkin Lymphomas (HL), a type of lymphoma, are characterized by the presence of Hodgkin–Reed–Stenberg (HRS) cells, which can be either mono- or multi-nucleated. HL can be further classified into classical HL (cHL) and nodular lymphocyte-predominant HL (NLPHL). cHL, which is the more common subtype (making up about 95% of HL cases), includes mixed cellularity, nodular sclerosis, and lymphocyte-rich subtypes. In contrast, NLPHL is a rare subtype, accounting for only 5% of HL cases, and is characterized by lymphocyte-predominant (LP) cells [19].

Inflammatory cells like T and B cells, tumor-associated macrophages (TAMs), mast cells, plasma cells, eosinophils, myeloid-derived suppressor cells, and NK cells are present in the HL tumor microenvironment and secrete cytokines and chemokines that control tumor angiogenesis, progression, and metastasis.

The key mediator of tumor angiogenesis is the VEGF. In HL, both HRS cells and TAMs secrete VEGF [20,21]. However, the lack of connection between VEGF expression and microvascular density suggests that other pro- and anti-angiogenic molecules, such as fibroblast growth factor-2 (FGF-2), hepatocyte growth factor (HGF), MMP-2, MMP-9, and hypoxia inducible factor 1 alpha (HIF-1), may also play a role in regulating angiogenesis in HL [17]. At least two different angiogenic processes appear to be involved in the promotion of lymphoma development and progression: paracrine effects of the proangiogenic tumor microenvironment and autocrine stimulation of tumor cells via production of VEGF and VEGFR by lymphoma cells. When compared to data from other solid tumors, there is little information available regarding the role of the HGF/c-MET signaling pathway in lymphomas. In B cell lymphoma cell lines and patient samples, however, there was no evidence of MET gene amplification.

Furthermore, it is unclear whether aggressive lymphomas are linked to high microvessel density. According to a group of scientists, microvascular density is greater in aggressive than indolent lymphomas as well as lymphomas compared to reactive nodes. However, it has been discovered that the microvascular density in reactive nodes is more or on par with that seen in lymphomas, particularly large cell lymphomas.

Korkolopoulou et al. demonstrated that in HL microvascular density is reduced with stage progression according to Ann Arbor stages I-IV [22]. Further research investigated the expression of HIF-1α in HL and found that it was present in HRS cells but did not correlate with increased microvascular density [23]. Another study focused on FGF cytokines and their receptors in HRS cells but did not find a direct correlation between their expression and the formation of new blood vessels [24]. In patients with HL, pre-treatment levels of VEGF and HGF were elevated but significantly reduced after therapy, and both pre- and post-treatment VEGF levels were found to be predictive of survival [25]. Furthermore, elevated serum VEGF levels in pre-treatment HL patients were reduced in cases of prolonged complete remission, but still remained higher compared to healthy individuals [26]. Dimtsas et al. evaluated the expression pattern of VEGF-A and VEGF receptor-1 and -2 (VEGFR-1 and VEGFR-2) in cHL and NLPHL and found that they were expressed in the HRS and lymphocytic and histiocytic cells [27].

## 4. Angiogenesis and Microvascular Density in Non-Hodgkin’s Lymphomas

B-cell lymphomas, including diffuse large B-cell lymphomas (DLBCLs), follicular lymphomas (FLs), extranodal marginal zone lymphomas, chronic lymphocytic leukemia (CLL), and mantle cell lymphomas (MCLs), represent 88% of all non-Hodgkin’s lymphomas (NHLs), while T and natural killer (NK) cell lymphomas 12%. T-cell lymphomas are more aggressive than B-cell lymphomas [24].

In Burkitt’s lymphoma and peripheral T-cell lymphoma (PTCL), microvascular density tends to be higher, while it tends to have intermediate values in DLBCL, and is lower FL [25]. In DLBCL, an increased vascular density (determined by the vascular maturation index, calculated as the ratio of LH39/CD34^+^ to all CD34^+^ vessels) has been demonstrated compared to that in FL [28]. Ultrastructurally, the stroma of B cell-NHLs contains immature vessels. These capillaries are made up of two endothelial cells that are parallel to one another and have thicker cytoplasm, creating a lumen that resembles a slit [25]. The distinguishing feature of follicular intermediate- and low-grade B-cell NHLs is the continuous basement membrane enclosing differentiated fenestrated capillaries. The blood vessels lumen in low-grade B-cell NHLs can develop in two different ways: either by curving the endothelial cell body or, more frequently, by the fusion of intracellular vacuoles in undifferentiated endothelial cells. High-grade B-cell NHLs, on the other hand, frequently have a distinct pattern of blood vessel growth that is defined by the creation of a slit-like lumen through neo-angiogenesis [27]. There is no known relationship between the microvascular density and the histologic subtype of NHLs [28].

Other studies on NHLs and DLBCL have found a correlation [29,30] or no correlation between microvascular density and VEGF expression [31,32,33]. Gratzinger et al. reported that the average microvascular densities significantly correlates with the intensity of VEGF staining [29]. Studies have shown that in both cutaneous T-cell and B-cell lymphomas, the microvascular density is higher compared to skin with a benign cutaneous lymphoid infiltrate [34,35,36]. Studies have shown that in both cutaneous T-cell and B-cell lymphomas, the microvascular density is higher compared to the skin with a benign cutaneous lymphoid infiltrate [37,38]. According to research, aggressive T cell lymphomas exhibit high levels of expression of VEGF-A compared to indolent B cell lymphomas [39]. This suggests that VEGF-A may play a role in the progression and aggressiveness of certain types of lymphoma. It is worth noting that while a minority of indolent follicular lymphomas (FLs) do show variable expressions of VEGF-A, it is not a consistent feature across all indolent B cell lymphomas [31,40]. VEGFRs expression levels are correlated with the level of VEGF expression in DLBCL [29]. CLL also expresses VEGFRs [41]. VEGF prevents apoptosis and increases the phosphorylation of VEGFRs. Immunocytochemical methods demonstrated the expression of VEGFRs, suggesting that VEGF transduction pathway is active in CLL [41].

HIF-1 and -2 and VEGF have a lower expression in indolent compared to aggressive lymphomas [41]. Indolent lymphomas transforming into aggressive lymphoma, express VEGF-A [42]. Angiogenesis is also correlated with mast cell density in B-NHL, according to the capacity of mast cells to release angiogenic factors [43].

## 5. Vasculogenic Mimicry and IMG in Lymphomas

Crivellato et al. demonstrated that in B-cell NHLs at the ultrastructural level, tumor cells are closely intermingled with endothelial cells and that this relationship can be recognized in the early stages of vessel formation, as an expression of vasculogenic mimicry [43]. Lymphoid tumor cells are indeed closely intermingled with vacuolated endothelial cells, and this relationship can be recognized in the early stages of vessel formation when immature endothelial cells have not yet formed a vascular lumen. Moreover, the tumor cells appeared to be completely enveloped by the cytoplasmic expansions of one or more endothelial cells, while vascular spaces were occasionally lined by lymphoid tumor cells. Moreover, Crivellato et al. showed that both low- and high-grade B-NHLs develop transluminal bridges in larger vessels, causing the parent vessel to split into two or more sections, suggesting that an intussusceptive modality of vascular growth also takes place in B-NHLs. This vascular pattern was more frequent in the center than on the margins of the lymphomas [43].

Using immunocytochemistry and confocal laser imaging, more proof of vasculogenic mimicry has been found in primary diffuse central nerve system lymphomas (PCNSL). Studies have demonstrated that a variety of cells, including CD20^+^ tumor cells, factor CD31^+^ endothelial cells, aquaporin-4 (AQP4)^+^ tumor cells, CD31^+^ endothelial cells, and CD20^+^ and AQP4^+^ tumor cells, engage in vessel formation [44]. PCNS B-cell lymphoma tumor cells show positivity with an anti-CD20 antibody. Blood–brain barrier stability and activity are associated with AQP4 expression, and this expression shifts in neurological conditions that disrupt the BBB. Peritumoral edema is linked with AQP4 expression, and AQP4 is significantly increased and redistributed over the borders of tumor cells in glioblastoma.

In 2003, Passalidou et al. showed that microvascular density was significantly greater in the paracortex than in the follicles in reactive lymph nodes and in FL. Interestingly, both reactive and neoplastic follicles did not significantly differ in microvascular density. In addition, the paracortex of reactive nodes showed higher microvascular density compared to FL and DLFL paracortex, demonstrating that tumor-induced angiogenesis is less effective than normal angiogenesis in responsive nodes [26]. Taken together, these findings suggest that vasculogenic mimicry and alternative modes of vascularization may play a significant role in the progression and survival of lymphomas and may contribute to the development of new therapeutic strategies targeting tumor angiogenesis.

## 6. Prognostic and Therapeutic Implications of Angiogenesis in Lymphomas and Alternative Mode of Vascular Growth as a Mechanism of Resistance to Anti-Angiogenic Therapies

Due to the variability of illnesses, various classifications, and research techniques, the prognostic and predictive significance of microvascular density and angiogenic variables in lymphomas is still debatable (immunohistochemistry, serum levels of angiogenic markers, mRNA extraction). Estimating microvascular density and VEGF are important for the development of NHL. Numerous investigations have investigated the connection between microvascular density, VEGF expression, and NHL prognosis; however, these studies have produced contradictory findings. Chemo resistant DLBCL and those with chemo sensitive lymphomas have different patterns of microvascular densities [41]. Progression-free (PFS) and overall survival (OS) are higher in FL patients treated with chemotherapy in conjunction with anti-angiogenic interferon-alpha2b, when the microvascular density is high before the treatment [40]. However, there was no association between elevated microvascular density and VEGF expression in individuals with DBLC after anthracycline treatment [45]. High serum VEGF levels before treatment have been proven to be prognostic indicators for survival in NHL [46]. However, it has been found that the pre-treatment serum level of VEGF is negatively correlated with both OS rate and disease-free survival in T and B cell lymphomas [47]. High serum VEGF levels have also been linked to unfavorable outcomes in DLBCL patients [48].

Furthermore, FGFR-1 expression correlates with lower frequencies of full remission in NHL patients, whereas FGF-2 expression is related with poor OS and PFS [49]. Moreover, Additionally, blood FGF-2 levels did not change following chemotherapy, nor was a connection established between microvascular density and the histological grade or prognosis [49]. High levels of FGF-2 before therapy have been found to independently predict survival, irrespective of other risk factors [46]. Soluble levels of VEGF, FGF-2, and PDGF-β declined after radiotherapy in NHL patients [50]. Blocking the VEGF-VEGFR pathway with neutralizing antibodies or tyrosine kinase inhibitors reduced p-STAT-3 levels and induce apoptosis in CLL [51]. High expression of both VEGF and VEGFR-1 in DBLC patients has been linked to increased OS and PFS following anthracycline therapy [45]. However, higher tissue expression of VEGF has been associated with unfavorable outcomes [52]. When compared to reactive lymph nodes, VEGF-A has been discovered to be overexpressed in tumor and endothelial cells in angioimmunoblastic T-cell lymphoma, and this overexpression is linked to a short survival time [53]. Increased VEGF expression is linked to aggressive DLBCL and subgroups of DLBCL with poor prognosis, as well as the transition from indolent B cell lymphoma [54]. VEGF expression in PCNSLs correlates with microvascular density and is linked to longer survival and changes to the blood–brain barrier [55].

An adaptive response to the conventional use of antitumor and antiangiogenic drugs to reestablish the normal characteristics of the vasculature, improving drug delivery and treatment efficacy, is represented by a transition from angiogenesis to IMG [56]. Sprouting angiogenesis is common in untreated tumors, but IMG is common following short-term therapy, restoring a vasculature with a modest rate of endothelial proliferation. A major path of acquired resistance to anti-angiogenic therapy is non-angiogenic growth. Using the existing vasculature and increasing the fraction of co-opted vessels, tumor cells may be able to resist anti-VEGF treatments [57]. One way of acquiring resistance to anti-VEGF treatments is vascular co-option [58,59,60].

## 7. Conclusions and Future Directions

The last literature evidence has clearly demonstrated that even if angiogenesis is a hallmark of cancer progression, an alternative mode of vascularization of tumor growth occurs, including IMG, vascular co-option, and vasculogenic mimicry, and in this context, tumors may have areas in which classic tumor angiogenesis occurs and other ones in which alternative vascularization takes place.

The advances in cancer biology due to the discovery of non-angiogenic growth have, therefore, the potential to lead to further steps toward more effective cancer treatments. In this context, a potential new therapeutic strategy when an angiogenic tumor treated with anti-angiogenic molecule “escape” by becoming non-angiogenic and not responding to anti-angiogenic treatment might be to combine anti-angiogenic chemicals with the blocking of vascular co-option.

## Figures and Tables

**Figure 1 cancers-15-03262-f001:**
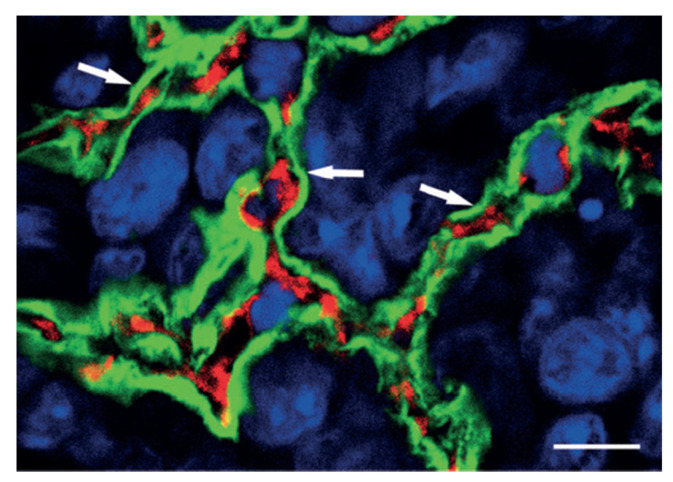
As indication of vascular co-option is the preservation of alveolar architecture in the outer regions of lung metastases. The tumor mass is highlighted in this high-power confocal image of the HT1080 lung metastasis that has been labeled with podoplanin (green fluorescence), CD31 (red fluorescence), and TOTO-3 (blue fluorescence). Note the intact alveolar walls with normal layering (pneumocyte-capillary-pneumocyte) (arrows). Scale bar = 10 µm. (Reprinted with permission from Ref. [9]).

**Figure 2 cancers-15-03262-f002:**
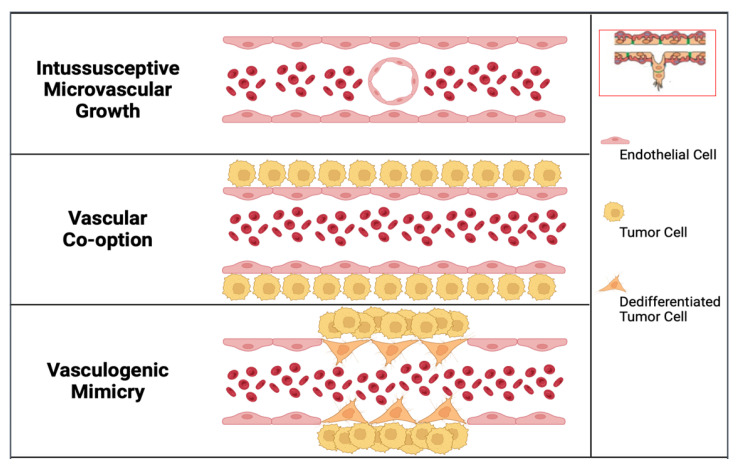
Alternative modes of tumor vascularization, compared with classic angiogenic mode (inset in the right corner of the figure). In tumor angiogenesis, through the local breakdown of the basement membrane of a pre-existing blood vessel, the underlying endothelial cells invade the surrounding stroma in response to an angiogenic cytokine. In intussusceptive microvascular growth, the expansion of the vascular network is achieved by inserting tissue columns into the vascular lumen of pre-existing vessels, allowing the formation of a new vessel by the splitting of the pre-existing one. In vascular co-option, invading cancer cells utilize pre-existing host vessels. In vasculogenic mimicry, cancer cells may be directly involved in the formation of blood vessels, intermingled with endothelial cells.

## Data Availability

The data presented in this study are available in this article.

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
