# Peer review of "Vascular Growth in Lymphomas: Angiogenesis and Alternative Ways"

_cancers, 2023, doi:10.3390/cancers15123262_

Round 1

Reviewer 1 Report (Previous Reviewer 2)

The authors have addressed my comments and the manuscript is more coherent now.

Author Response

We acknowledge the Reviewer 1 for his/her positive evaluation.

Reviewer 2 Report (Previous Reviewer 1)

The authors of the manuscript 'The alternative mode of vascular grothe in lymphomas' did substancial rewriting and giva a point to point reply to both reviwers. Yet the results of these efforts are not sufficient. 

The title does not reflect the content. Tumor angiogenesis/tumor growth is explained not fully understaning Folkman's concept. Tumor dormany is neglected.

References are not adequately given (for each statement). Recent literature for angiogenesis is missing. Sometimes summary statements are given without naming the authors. 

It is still unlcear, how Figure 1 is relating to the topic (title is lymphomas), even though the legend improved. HT1080 remains a sarcoma.

Figure 2 is a starting point, but need major improvement. Much prior knowledge would be necessary, since explanations are missing. Camparison to angiogenesis is missing.

 Studies and molecules are mainly listed. Even though better definitions of terms like microvasular density, immature vessels etc. have been provided, the use of key markers need more in depth discussion. This includes CD31 vs CD34, LH39 vs the more common SMA.

Given the conflicting if not confusing discription of numerous studies, it is unclear what leads the authors to state that 'angiogenesis is not a hallmark of cancer'.

The conclusions leave this reviewer confused. Vasularisation is not evenly distributed within the tumor (this IS a characteristic of tumors). The terms tumor vasculature and angiogenesis are intermixed. Tumor cell migration and tumor growth are depicted in a very simplyfied way. 

English language is mainly correct.

Author Response

We acknoledge the Reviewer 2 for his/her comments.

In reply:

The title does not reflect the content. Tumor angiogenesis/tumor growth is explained not fully understanding Folkman's concept. Tumor dormancy is neglected.

The actual title of the MS “The alternative mode of vascular growth in lymphomas” concerns the alternative models of tumor vascularization involved in lymphoma described in the text, not related to tumor angiogenesis according to Folkman’s concept. In 2018, I have written a biography of Judah Folkman (Judah Folkman. A biography. DORDRECHT, Springer, 2018. ISBN 978-3-319-92632-2) and I know very well the “paradigm” which inspired the work of Folkman.

I have explained the concept of tumor dormancy as follows: “Another important aspect of tumor growth is represented by tumor dormancy, occurring during tumor initiation (“local tumor dormancy”), metastatic dissemination (“metastatic dormancy”), and escape from anti-cancer therapies (“therapy-induced dormancy”). Tumor dormancy can be also distinguished in the arrest of cancer cell proliferation within the tumor mass caused by apoptosis, due to poor vascularization (“angiogenic dormancy”), or by the immune response (“immune dormancy”)( Pranzini E, Raugei, Taddei ML, 2022. Metabolic features of tumor dormancy: possible therapeutic strategies. Cancers,14: 547).

References are not adequately given (for each statement). Recent literature for angiogenesis is missing. Sometimes summary statements are given without naming the authors. 

I have improved the number of references according to Reviewer’s suggestion.

It is still unclear, how Figure 1 is relating to the topic (title is lymphomas), even though the legend improved. HT1080 remains a sarcoma.

The Figure 1 reproduces one of the first examples in literature concerning vascular co-option. There are not still examples of vascular co-option in lymphomas.

Figure 2 is a starting point but need major improvement. Much prior knowledge would be necessary since explanations are missing. Comparison to angiogenesis is missing.

We have modified the Figure 2 by adding an inset concerning tumor angiogenesis and we have also modified the legend to the Figure as follows: “Alternative modes of tumor vascularization, compared with classic angiogenic mode (inset). In tumor angiogenesis, through the local breakdown of the basement membrane of a pre-existing blood vessel, the underlying endothelial cells invade the surrounding stroma in response to an angiogenic cytokine. In intussusceptive microvascular growth, the expansion of the vascular network is achieved by inserting tissue columns into the vascular lumen of pre-existing vessels, allowing to the formation of a new vessel by the splitting of the pre-existing one. In vascular co-option, invading cancer cells utilize pre-existing host vessels. In vasculogenic mimicry, cancer cells may be directly involved in the formation of blood vessels, intermingled with endothelial cells.

 Studies and molecules are mainly listed. Even though better definitions of terms like microvascular density, immature vessels etc. have been provided, the use of key markers needs more in depth discussion. This includes CD31 vs CD34, LH39 vs the more common SMA.

We have specified that CD31 is more expressed in mature endothelial cells, whereas CD34 is more expressed in immature blood vessels. In the meantime, as concerns endothelial basement membrane, LH39 is a monoclonal antibody recognizing an epitope located at the lamina lucida of mature small veins and capillaries but not in newly formed vessels (Almeida BM, Challacombe SJ, Eveson JW, Morgan PR, Purkis PE and Leigh IM (1992) The distribution of LH39 basement membrane epitope in the tumour stroma of oral squamous cell carcinomas. J Pathol 166: 369–374) 

Given the conflicting if not confusing description of numerous studies, it is unclear what leads the authors to state that 'angiogenesis is not a hallmark of cancer'.

We have better explained the concept as follows: “The last literature evidence has clearly demonstrated that even if angiogenesis is a hallmark of cancer progression, also an alternative mode of vascularization of tumor growth occurs, including IMG, vascular co-option, and vasculogenic mimicry, and in this context, tumors may have areas in which classic tumor angiogenesis occurs, and other ones in which alternative vascularization takes place.

The conclusions leave this reviewer confused. Vascularization is not evenly distributed within the tumor (this IS a characteristic of tumors). The terms tumor vasculature and angiogenesis are intermixed. Tumor cell migration and tumor growth are depicted in a very simplified way. 

We have modified the conclusions as follows: “The advances in cancer biology due to the discovery of non-angiogenic growth have therefore the potential to lead to further steps toward a more effective cancer treatment. In this context, a potential new therapeutic strategy when an angiogenic tumor treated with anti-angiogenic molecule from "escape" by becoming non-angiogenic and no more responding to anti-angiogenic treatment, might be to combine anti-angiogenic chemicals with blocking of vascular co-option.”

Round 2

Reviewer 2 Report (Previous Reviewer 1)

With this second revision, not all points raised by this reviewer have been adressed. This includes the title, Fig, 1 (HT1080 still not beeing a lymphoma) and Fig. 2 lacking a comparison to 'angiogenesis'.   Yet the manuscript on alternative modes of tumor vascularization again has been improved. 

Author Response

We have changed the title as follows “Vascular growth in lymphomas: angiogenesis and alternative ways; concerns Figure 1, as we have specified in the previous report the reproduces one of the first examples in the literature concerning vascular co-option, and it is not referred to lymphoma, because there are not still reports of vascular co-option in lymphomas; Figure 2 now includes a comparison to angiogenesis by adding an inset in the right corner of the figure showing a schema of concerning tumor angiogenesis and we have also modified the legend to the Figure as follows: “Alternative modes of tumor vascularization, compared with classic angiogenic mode (inset). In tumor angiogenesis, through the local breakdown of the basement membrane of a pre-existing blood vessel, the underlying endothelial cells invade the surrounding stroma in response to an angiogenic cytokine. In intussusceptive microvascular growth, the expansion of the vascular network is achieved by inserting tissue columns into the vascular lumen of pre-existing vessels, allowing to the formation of a new vessel by splitting the pre-existing one. In vascular co-option, invading cancer cells utilize pre-existing host vessels. In vasculogenic mimicry, cancer cells may be directly involved in the formation of blood vessels, intermingled with endothelial cells.

This manuscript is a resubmission of an earlier submission. The following is a list of the peer review reports and author responses from that submission.

Round 1

Reviewer 1 Report

The manuscript ‚The alternative mode of vascular growth in lymphomas‘ reviews modes of tumor vessel formation. The different types of lymphomas and their vascularisation are described.

Yet the title does not reflect major aspects of the manuscript, in which chapters 2 and 3 specifically deal with angiogenesis.

It seems that sometimes the terms angiogenesis and vasculogenis are mixed. A clear definition of all related terms (e. g. in a table) would enhance the manuscript.

The two figures show very specific examples of vascular structures. For Figure 1 markers should be better decribed, the morphological structure is unclear to readers not familiar with the tumor model. In addition HT1080 is a fibrosarcoma cellline – and lung metastasis suggest a murine model.

Figure 2 uses a different marker (CD34 insead of CD31) for vasularisation. The  resolution/magnification is to low, to identify the type of vascular structures. A morphometric analysis is shown, but it is unclear how CD34 positivity is defined. Area of CD34 positivity alone does not reflect the type of vascularisation (e. g. vasuclar mimicry).

The types of tumor vascularisation are described in the text, a schematic or tabular depiction would enhance the manuscript.

Some signalling molecules are listed, but the underlying signalling cascades should be explained in more depth. E. g. HGF is only refered to as ‚another growth factor‘).

Microvascular density should be defined in more detail (e. g. markers, measurement strategy).

The term ‚immature vessels‘ should be explained in more detail.

In summary this manuscript deals with an important aspect of lymphoma tumorigenesis, includes specifics of lymphoby subtypes, but needs some improvements.

Reviewer 2 Report

The alternative mode of vascular growth in lymphomas 

Interestingly, an alternative method for angiogenesis has been proposed for lymphoma.

This paper is a review but has been presented as a communication.

Lines 41-42 talk about the 3 alternative modes of angiogenesis; IMG, vascular co-option and vasculogenic mimicry and these have then been defined. I would suggest that the authors add a representative figure of IMG to display (inserting tissue columns into the vascular lumen of pre-existing vessels), in addition, this can be done for co-option and mimicry.

In line 46 it is unclear if the three paths described earlier apply to two main alternative mechanisms of vascularisation. Is this where the authors introduce co-option and mimicry or are these separate? I’m not sure.

Figure 1 explains co-option, so kindly make this clearer in the paragraph inclusive of 46-52.

I am not sure why the authors have given this example (figure 1) since none of the markers shown in the figure has been discussed before this and the authors assume prior knowledge of the readers.

In the paragraph commencing with line 64, co-opting has been added. What is meant by cuffs in line 65? The authors need to explain why co-opting leads to blood vessel apoptosis. Also, I don’t understand the logic of then following up this sentence with the role of VEGF. In general, the logical order of topics is not great.

In the next line, mimicry is mentioned but it is not explained well. How does the involvement of the formation of blood vessels lead to mimicry? The mosaic sentence does make sense.

Neoplastic cells and macrophages express vascular endothelial growth factor (VEGF) in HL. This is interesting that macrophages can also contribute to angiogenesis.

Again, the topic abruptly shifts:

The absence of a significant relationship between VEGF expression and microvascular density implies that VEGF may not be the only factor regulating angiogenesis in HL. How is this linked to the preceding sentence about the role of macrophages? Also, microvascular density suddenly appears with no apparent explanation of what it means. Again, prior knowledge is assumed.

Microvascular density is finally introduced in line 93. 

Specifically, they looked at the expression of VEGF, MMP-2, MMP-9, TIMP1, HIF-1α, and FGF cytokines and their receptors in HRS cells [19]. However, these studies found that microvascular density (the number of blood vessels in a given area) was not correlated with the expression of these factors in HRS cells. Further research investigated the expression of HIF-1α in HL and found that it was present in HRS cells but did not correlate with increased microvascular density. What is the point the authors are trying to make here, what factor was finally correlated with microvascular density in HRS?

Interestingly, pre-and post-treatment VEGF levels were found to predict survival. Can the authors be more specific, are these increased or decreased levels and increased or decreased survival?

The issue with the writing style of this manuscript is that a few vague sentences have been provided about each added study and on some occasions the point the authors are trying to make is unclear or it is not clear in what direction the flow of topics is going.

Some introductory information was provided about NHL. Some comparative comments about microvasculature in different types of B-cell NHL. This is also superficial and dispersive.

Again, the markers in Figure 2 have not been introduced, nor has it been referred to in the main text at all. I don’t see what it is adding to the paper.

Microvascular density is then compared amongst B cell NHL. I am totally confused about the topic of this manuscript. I initially thought the three main alternative vascularisation paths were intended and now the focus seems to have shifted to density.

Research suggests that in DLBCL, the degree of VEGF expression is correlated with the expression levels of VEGFR-1 and VEGFR-2 [30]. Additionally, VEGFR-1, VEGFR-2, and VEGFR-3 are also expressed in Chronic Lymphocytic Leukemia (CLL) [42]. VEGF has been shown to prevent apoptosis and increase the phosphorylation of VEGFR-1 and VEGFR-2.  Some content has been added to the input of VEGF. Also, the self-stimulating pro-survival loop in CLL needs to be explained more in relation to the paragraph above.

Section 4 talks about mimicry in B cell NHL.

The studies have shown that different cells participate in vessel formation, including CD20+ tumor cells and factor VIII+ endothelial cells, aquaporin-4 (AQP4)+ tumor cells and CD31+ endothelial cells, as well as CD20+ and AQP4+ tumor cells. I am not sure what the mention of these factors (some very well-known such as CD31) adds at all when it is not explained in any depth.

After this, the macrovascular topic ensues but co-option has not been explained in any depth.

Section 5,

A switch from angiogenesis to IMG represents an adaptive response to treatment 199 with antitumor and antiangiogenic compounds to restore the hemodynamic and morphological characteristics of the vasculature, favouring tumor drug delivery and sensitivity to treatments. How is this relevant? Kindly explain it more.

Then the topic shifts again to microvascular density. Drugs mentioned in this section just pop out of the blue, kindly mention whether they are linked to vascularisation. Then again a shift of topic happened to VEGF and density.

In addition, Pazgal et al. demonstrated that in patients with NHL, FGF-2 expression is associated with poor overall and progression-free survival, whereas FGFR-1 expression correlates with reduced rates of complete remission [52].  This is better. The final paragraph linking treatment and VEGF levels with the outcome is useful.

In all fairness and honesty, the logical flow of the topics in this manuscript is not great, paragraphs are fragmented and topics rapidly shift in and out of focus. The figures have not been explained and the authors assume prior knowledge.